# Mixed-Precision Inference Quantization: Problem Resetting and traditional NP hard problem

## Abstract

Based on the model's resilience to computational noise, model quantization is important for compressing models and improving computing speed. Existing quantization techniques rely heavily on experience and "fine-tuning" skills. In this paper, we map the mixed-precision layout problem into a traditional NP hard problem and the problem can be solved by low cost methods like branch and bound method without "fine-tuning". In experiments, experimental results show that our method is better than HAWQ-v2, which is one of the current SOTA methods to solve mixed-precisions layout problem.

## 1 Introduction

Neural network storage, inference, and training are computationally intensive due to the massive parameter size of neural networks. Therefore, developing a compression algorithm for machine learning models is necessary. Model quantization, based on the robustness of computational noise, is one of the most important compression techniques. The computational noise robustness measures the algorithm's performance when noise is added during the computation process. The primary sources of noise are truncation and data type conversion mistakes.

The initial high-precision data type used for a model's parameters is replaced with a lower-precision data type during model quantization. It is typical to replace FP32 with FP16, and both PyTorch and TensorFlow have quantization techniques that map floats to integers. Various quantization methods share the same theoretical foundation, which is the substitution of approximation data for the original data in the storage and inference processes. A lower-precision data format requires less memory, and using lower-precision data requires fewer computer resources and less time. In quantization, the precision loss in different quantization level conversions and data type conversions is the source of the noise.

The primary issue with the model quantization methods is that a naive quantization scheme is likely to raise the loss function. It is not easy to substitute massive-scale model parameters with extremely low-data precision without sacrificing significant precision. It is also not possible to utilize the same quantization level, i.e., to introduce the same level of noise to all parameters for all model parameters and get good performance.

Utilizing mixed-precision quantization is one way to solve this issue. For more "sensitive" model parameters, higher-precision data is used, whereas lower-precision data is used for "nonsensitive" model parameters. Higher-precision data indicates that the original data adds small noise, while lower-precision data indicates that the original data adds large noise. But mixed-precision quantization also has its restrictions: In computing processes, for example, GPU and CPU must use the same data type in a computing process.

Moreover, the following facts challenge current algorithms for mixed-precision algorithms: 1. These algorithms are built on empirical experience and "tuning" skills. 2.Some algorithms forego neural network and dataset analysis. Some algorithms base model quantization on hardware features. It is impossible to show the bounds of algorithms' performance. 3. Some algorithms utilize Hessian data. The majority of them are analyzable. However, obtaining Hessian information necessitates

a considerable computing resources and time. Some of these methods are only useful for storing purposes.

The consensus among researchers is that quantization technology without "fine-tuning" is harmful to the performance of the model. Current quantization technology widely uses the "fine-tuning" method, but no one can explain why their algorithm cannot work without "fine-tuning", even though the basis of their algorithm is well-defined in math.

This paper establishes the model quantization problem setting for inference processes. Based on our analysis, we map layerwise PTQ problem into one of the traditional NP-hard problems, i.e.,extended 0-1 knapsack problems, which can be solved by the branch and bound method without too much computing resources or inference/training model too many times. Thus, our work gets rid of "fine-tuning" skills and has clear interpretability. Compared with the SOTA mixed-precision algorithm HAWQ-v2 without "fine-tuning" the quantization model based on our work is better than the model from HAWQ-v2 under the same computation resource limitation.

## 2 RELATED WORK

Model compression methods include pruning methodsHan et al. (2015); Li et al. (2016); Mao et al. (2017) , knowledge distillationHinton et al. (2015), weight sharingUllrich et al. (2017) and quantization methods. From the perspective of the precision layout, post-training quantization methods can be mainly divided into channelwise Li et al. (2019); Qian et al. (2020), groupwise Dong et al. (2019b) and layerwise Dong et al. (2019a) methods. Layerwise mixed-precision layout schemes are more friendly to hardware. Parameters of the same precision are organized together, making full of a program's temporal and spatial locality. A common problem definition for quantizationDong et al. (2019a); Morgan et al. (1991); Courbariaux et al. (2015); Yao et al. (2020) is as follows Gholami et al. (2021).

**Problem 1** *The objective of quantization is to solve the following optimization problem:*

$$\min_{q \in \mathbf{Q}} \|q(w) - w\|^2$$

*where $q$ is the quantization scheme, $q(w)$ is the quantized model with quantization $q$, and $w$ represents the weights, i.e., parameters, in the neural network.*

Although problem 1 gives researchers a target to aim for when performing quantization, the current problem definition has two shortcomings: 1. The search space of all possible mixed-precision layout schemes is a discrete space that is exponentially large in the number of layers. There is no effective method to solve the corresponding search problem. 2. There is a gap between the problem target and the final task target. As we can see, no terms related to the final task target, such as the loss function or accuracy, appear in the current problem definition.

## 3 BACKGROUND ANALYSIS

### 3.1 MODEL COMPUTATION, NOISE GENERATION AND QUANTIZATION

Compressed models for the inference process are computed using different methods depending on the hardware, programming methods and deep learning framework. All of these methods introduce noise into the computing process. One reason for this noise problem is that although it is common practice to store and compute model parameters directly using different data types, only data of the same precision can be support precise computations. Therefore, before performing computations on nonuniform data, a computer will convert them into the same data type. Usually, a lower-precision data type in a standard computing environment will be converted into a higher-precision data type; this ensures that the results are correct but require more computational resources and time. However, to accelerate the computing speed, some works on artificial intelligence (AI) computations propose converting higher-precision data types into lower-precision data types based on the premise that AI models are not sensitive to compression noise. The commonly used quantization technology is converting data directly and using a lower-precision data type to map to a higher-precision data type linearly.

We use the following example to illustrate quantization method, which is presented in Yao et al. (2020). Suppose that there are two data objects $input_1$ and $input_2$ are to be subjected to a computing operation, such as multiplication. After the quantization process, we have $Q_1 = \text{int}(\frac{input_1}{scale_1})$ and $Q_2 = \text{int}(\frac{input_2}{scale_2})$, and we can write

$$Q_{output} = \text{int}(\frac{input_1 * input_2}{scale_{output}}) \approx \text{int}(Q_1 Q_2 \frac{scale_1 * scale_2}{scale_{output}})$$

$scale_{output}$, $scale_1$ and $scale_2$ are precalculated scale factors that depend on the distributions of $input_1$, $input_2$ and the output; $Q_i$ is stored as a lower-precision data type, such as an integer. All $scale$ terms can be precalculated and established ahead of time. Then, throughout the whole inference process, only computations on the $Q_i$ values are needed, which are fast. In this method, the noise is introduced in the $\text{int}(\cdot)$ process. This basic idea gives rise to several variants, such as (non)uniform quantization and (non)symmetrical quantization.

When we focus on quantization strategy, i.e. $round$ function in quantization framework like Micronet, we can have at least three strategy: round up, i.e., $ceil$ function in python, round down, i.e., $floor$ function in python and rounding, i.e., $round$ function in python. usually, rounding is the most common method to deal with quantization. But, in this paper, we will show that how to mixed use round up/round down to gain a mixed precision quantized model which is better than full precision model.

## 3.2 NEURAL NETWORKS

In this paper, we mainly use the mathematical properties of extreme points to analyze quantization methods. This approach is universal to all cases, not only neural networks. However, there is a myth in the community that it is the neural network properties that guarantee the success of quantization methodsWang et al. (2019); Morgan et al. (1991); Demidovskij & Smirnov (2020). To show that the properties of the extreme points, not the properties of the neural network, are what determine the ability to quantize, i.e. the ability to handle noise, we must first define what a neural network is.

The traditional definition of a neural network Denilson & Barbosa (2016) as a human brain simulation is ambiguous; it is not a rigorous mathematical concept and cannot offer any analyzable information. The traditional descriptions of neural networks Denilson & Barbosa (2016) focus on the inner products of the network weights and inputs, the activation functions and directed acyclic graphs. However, with the development of deep learning, although most neural networks still consist of weighted connections and activation layers, many neural networks no longer obey these rules, such as the network architectures for position embedding and layer norm operations in Transformers. Moreover, current deep learning frameworks, such as PyTorch and TensorFlow, offer application programming interfaces (APIs) to implement any function in a layer. Therefore, we propose that the definition of a neural network adheres to the engineering concept indicated by the definition 1 rather than a precise mathematical definition; that is, a neural network is a way for implementing a function.

**Definition 1** *The neural network is the function which is implemented in composite function form.*

A neural network can be described in the following Eq. 1 form.
$$model(x) = h_1(h_{2,1}(h_{3,1}(...), ..., h_{3,k}, w_{2,1}),$$
$$h_{2,2}(h_{3,k+1}(...), ..., w_3), ..., w_{2,2}), ..., w_1) \tag{1}$$
where $h_{i,j}$, $i \in [2, ..., n]$, are the $(n-i+1)$th layers in the neural network; $w_{i,j}$ is the parameter in $h_{i,j}(\cdot)$.

Definition 1 means that a neural network, without training, can be any function. With definition 1, a neural network is no longer a mathematical concept, and this idea is widely used in practice Roesch et al. (2019). We can see from definition 1 that the requirement that a neural network is in composite function form is the only mathematical property of a neural network that can be used for analysis.

In practice, the loss function is one method to evaluate a neural network. A lower loss on a dataset means a better performance neural network. For example, the training process optimises the model's loss, i.e., following Eq. 2.
$$\min_w f(w) = \mathbf{E}_{sample} \ell(w, sample) = \frac{1}{m} \sum_{(x_i, y_i) \in \mathbb{D}} \ell(w, x_i, y_i) \tag{2}$$

where $f(\cdot)$ is the loss for model on a dataset, $w$ represents the model parameters, $\mathbb{D}$ is the dataset, $m$ is the size of the dataset, $\ell(\cdot)$ is the loss function for a sample and $(x_i, y_i)$ represents a sample in the dataset and its label.

In this paper, we mainly use the sequential neural network to describe the conclusion for the sequential neural network is easily described, and the whole conclusion is non-related to the structure of the neural network. For a sequential $n$-layer neural network, $\ell(\cdot)$ can be described in the following Eq.3 form.

$$\ell(w, x_i, y_i) = L(model_n(x_i, w), y_i)$$
$$model_n = h_1(h_2(h_3(h_4(\cdots h_n(h_{n+1}, w_n) \cdots, w_4), w_3), w_2), w_1) \tag{3}$$

where $L(\cdot)$ is the loss function, such as the cross-entropy function; $h_i$, $i \in [1, ..., n]$, is the $(n-i+1)$th layer in the neural network; $w = (w_n^T, w_{n-1}^T, \cdots, w_1^T)^T$, $w_i$ is the parameter in $h_i(\cdot)$; and for a unified format, $h_{n+1}$ stands for the sample $x$.

## 4 START POINT AND THEORY PROBLEM SETTING

### 4.1 QUANTIZATION TARGET

The problem 1 cannot expose any useful information. Thus, we have to setting another quantization target. Because of the natural of quantization is the trade of the performance and the computation resource. From the current work, we know that generally, the less computation resource use, the worse the model's performance is. We want to give the model that has best quantization performance with limited computation resource, like limited memory cost. The performance of model can be measured by loss function. Thus, using memory cost as computational resource example. we have following problem setting.

**Problem 2**

$$\min \|f(w) - \bar{f}(w)\|, s.t. \|w\| < C \tag{4}$$

*where $C$ is the limitation of memory cost, $\bar{f}(w)$ is the loss of model after quantization.*

Again, problem 2 setting also cannot be solved. We need more description about $\bar{f}$.

### 4.2 ANALYSIS BASE

Quantization methods for inference are complex. In addition to the noise added to the parameters directly, noise is also introduced between different layers in the inference process because different quantization levels or data types of different precisions are used in different layers, which shown in figure 1.

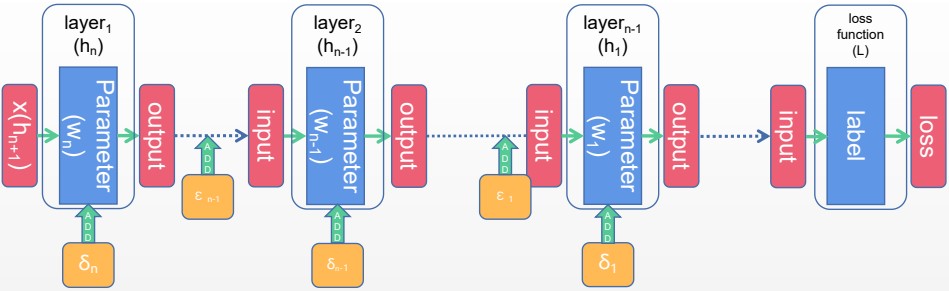

Figure 1: The quantization of models for use in the inference process. When quantized models are used in the inference process, the outputs of different layers suffer from noise due to the conversion between the different quantization levels of different layers in layerwise quantized models.

After quantization, the quantized loss for a sample, i.e. $\bar{\ell}(\cdot)$, in the inference process is as follows.

$$\bar{\ell}(w, x_i, y_i) = L(h_1(h_2(\cdots h_n(h_{n+1} + \epsilon_n, w_n + \delta_n) + \epsilon_{n-1} \cdots, w_2 + \delta_2) + \epsilon_1, w_1 + \delta_1), y_i)$$

where $\delta_i$, $i \in 1, \cdots, n$, and $\epsilon_i$, $i \in [1, ..., n]$, are the minor errors that are introduced in model parameter quantization and in data type conversion in the mixed-precision layout scheme, respectively.

Thus, we obtain the following expression based on the basic total differential calculation.

$$\bar{\ell}(w, x_i, y_i) - \ell(w, x_i, y_i) = \sum_{i=1}^{n} \frac{\partial \ell}{\partial h_{i+1}} \cdot \epsilon_i + \frac{\partial \ell}{\partial w_i} \cdot \delta_i \tag{5}$$

where $\cdot$ is inner product and $*$ is the scalar product in following parts. For the loss on whole dataset, we can gain

$$\min_{\epsilon \in E} \bar{f}(w) - f(w) = \frac{1}{m} \sum_{(x_j, y_j) \in \mathbb{D}} \sum_{i=1}^{n} \frac{\partial \ell}{\partial h_{i+1}} \cdot \epsilon_i + \frac{\partial \ell}{\partial w_i} \cdot \delta_i$$

$$= \frac{1}{m} \sum_{i=1}^{n} \sum_{(x_j, y_j) \in \mathbb{D}} \frac{\partial \ell}{\partial h_{i+1}} \cdot \epsilon_i \leq \frac{1}{m} \sum_{i=1}^{n} \left\| \sum_{(x_j, y_j) \in \mathbb{D}} \frac{\partial \ell}{\partial h_{i+1}} \epsilon_i \right\| \leq \frac{1}{m} \sum_{i=1}^{n} \left\| \sum_{(x_j, y_j) \in \mathbb{D}} \frac{\partial \ell}{\partial h_{i+1}} \right\| \|\epsilon_i\|$$

$$\tag{6}$$

where $\bar{f}(w) = \frac{1}{m} \sum \bar{\ell}(\cdot)$. The reason for second equation in Eq. 6 is for a well-trained model, the expectation of $\ell(\cdot)$'s gradient for parameters is zero, i.e., for the $\sum_{(x_j, y_j) \in \mathbb{D}} \frac{\partial \ell}{\partial w}$ components, $\frac{\partial \ell}{\partial w_i} = 0$.

As we can see, current works mixed discussed the quantization for storage and inference Dong et al. (2019a); Yao et al. (2020); Dong et al. (2019b); Nahshan et al. (2021). Consequently, these works must add a "fine-tuning" process, and they still fail in some cases. Moreover, this is why channel-wise quantization methods are booming. In a channel that uses the same data type at all times, the precision loss of the corresponding layer input is usually zero.

A frequently asked question is why $\sum_{(x_j, y_j) \in \mathbb{D}} \frac{\partial \ell}{\partial w}$ is zero but $\sum_{(x_i, y_i) \in \mathbb{D}} \frac{\partial \ell}{\partial h_{i+1}}$ is non-zero. The optimization algorithm is to optimize $w$ in the training process. Thus, $\sum_{(x_i, y_i) \in \mathbb{D}} \frac{\partial \ell}{\partial h_{i+1}}$ is random in the final model except for the layers with bias terms like the batch norm layer. The bias term will absorb the gradient and train them in the optimization process. What is more, in the model, which mainly consists of identity mapping, $\sum_{(x_i, y_i) \in \mathbb{D}} \frac{\partial \ell}{\partial h_{i+1}}$ is close to zero vector.

## 4.3 THEORY PROBLEM SETTING

Eq. 6 makes it extremely clear that the noise supplied from the output of the previous layer to the input of the later layer in a neural network exerts a first-order effect on the change of the final loss function, whereas noise on weights exerts a second-order effect on the loss function. In the quantization process, the input and output of the layer should take precedence over the quantization method, while the quantization of the weights should accommodate the impact brought by the quantization between layers.

When the gradient between layers, i.e., $\frac{\partial \ell}{\partial h_i}$ is large, the input and the weight of that layer should be quantized with a more precise approach, i.e., more bit data type. In this case, although it is possible that weight quantization in this layer does not require a high-bit data type, for example, the distribution of weight is a binary distribution, it cannot be stored with a simple boolean data type due to the significant error generated by the input layer quantization. When the gradient between layers is small, we can select a data type with the lowest possible precision to boost the speed of computing while still satisfying the mathematical concept of "neighborhood" for the magnitude of the noise introduced in both the layer weights and the layer inputs.

Based on the above description, we can derive the main idea of the algorithm: we want to solve the following problem by using total differential as the benchmark and prediction function, adjusting the quantization of $\frac{\partial \ell}{\partial h_i}$ in a mix-precision quantization scheme, and quantizing weight to accommodate the quantization of corresponding $\frac{\partial \ell}{\partial h_i}$. We modified problem 2 into the problem 3.

**Problem 3**

$$\min \sum_{i=1}^{n} \left\| \sum_{(x_j,y_j) \in \mathbb{D}} \frac{\partial \ell}{\partial h_{i+1}} \right\| \|\epsilon_i\| \, or \sum_{i=1}^{n} \left\| \sum_{(x_j,y_j) \in \mathbb{D}} \frac{\partial \ell}{\partial h_{i+1}} \epsilon_i \right\|, s.t. \|w\| < C \quad (7)$$

*where $C$ is the limitation of memory cost.*

Our problem setting for quantization is different from previous work like HAWQDong et al. (2019a); Yao et al. (2020); Dong et al. (2019b); Nagel et al. (2020) because these methods do not take the error in the layer's input into consideration, which prevents their work and analysis in the mixed-precision computing area. As a result, these works can only be used to store a compressed neural network on a disk. When the compressed model is stored in memory for inference, these compressed models have to be recovered into the full precision model.

# 5 MAP MATHEMATICAL CONCEPT INTO PRACTICAL OPERATION AND REAL PROBLEM SETTING

Although we have a target problem 3, yet all variable in Eq. 7 is purity mathematical concept and cannot be used to indicate the practical operation like quantization level choosing. Thus, we have to map mathematical concept into practical operation

## 5.1 MAP NOISE INTO QUANTIZATION LEVEL

Before using Eq. 6 to design an algorithm, we have to deal with the problem that maps the math concepts into practical operations and concepts that can be implemented by a computation device.

Based on the preceding discussion, the first consideration when mapping from the concept of noise to the quantization level or data type is how to describe the noise into a data type and how to design a method in which noise can be covered by the mathematical concept of "neighborhood". And if the value of noise exceeds the mathematical concept of "neighborhood", it must be adjusted based on the practical environment, computation source, and target.

### 5.1.1 MAP $\epsilon$ INTO QUANTIZATION METHOD PARAMETERS

Based on the above explanation, the fluctuations due to quantization are almost a linear function of the $\epsilon$. Thus in order to restrict the changes of the loss function caused by quantization, the $\epsilon$ vector should have the smallest second norm value, i.e., the quantization parameter setting's objective is $\min \epsilon^2$. On the basis of the above objective function, it is simple to choose optimal parameter configurations for many quantization schemes.

Using symmetric and asymmetric quantization schemes as examples in this paper, we need to set a clip value for the (a)symmetric INT quantization scheme. When the value is either less than $clip^-$ or greater than $clip^+$, we treat them as $clip^{(+/-)}$. Using the right clip setting can help narrow the range of data distribution and improve the accuracy of INT expression. Thus, the values of $clip^{(+/-)}$ parameters are very important. In this part, we will show why the ACIQ quantization methodBanner et al. (2019) is reasonable which is a experimental results currently.

The traditional approach is to use the KL divergence to determine the clip value. The problem is that the approximation of the KL divergence is not an approximation in the aspect of the norm. It only represents the difference between the two distributions in terms of information theory. But the difference in information theory cannot be directly reflected in the change of the loss function. In general, the mathematical property that directly reflects the change in the loss function is the loss function's gradient or the value that is relative to the gradient.

**Simple proof of ACIQ** Because of the Eq. 6, the fluctuation of loss function is related to the $\|\frac{\partial \ell}{\partial h_i}\|$ and $\|\epsilon_i\|$. $\|\frac{\partial \ell}{\partial h_i}\|$ is constant number for a network on a dataset. Thus, we have to minimize the $\|\epsilon\|$, i.e., $\|\epsilon\|^2$. Thus, when we use asymmetric quantization, then we should choose a clip value that satisfies the following conditions. When the data distribution is $p(x)$, and quantization datatype has

$B$ bits, the objective function is as following.

$$\min \|\epsilon\|^2$$

$$=> \min \int_{-\infty}^{clip^-} (x - clip^-)^2 p(x) dx + \int_{clip^-}^{clip^+} (x - \lfloor \frac{x * 2^B}{clip^+ - clip^-} \rfloor)^2 p(x) dx + \int_{clip^+}^{\infty} (x - clip^+)^2 p(x) dx$$

The above problem can be solved by direct derivation after determining the form of p(x), which is the content of the workBanner et al. (2019).

### 5.1.2 MAP $\frac{\partial \ell}{\partial h_i}$ INTO THE SLOPE OF SECANT LINE

In our discussion, a prerequisite for the correct application of total differentiation is that the noise that we introduce must be small enough to permit error estimation using total differentiation. However, in practice, we always expect faster computation by employing coarser data storage methods and lower-bit data types. In practice, there may be instances where large noise is used in the quantization process, such as quantization with INT4 or 2-value, 3-value on a vector with a wide value distribution. In our experiments with ResNet8 and 14 on the cifar10 dataset, the noise when quantization using the above quantization scales is roughly in the range of 0.1 to 0.3, indicating that the noise is outside the range that can be expressed in terms of "neighborhood".

Higher order differentials are useful for solving precise prediction problems, but a first-order approximation is more practical due to computational and memory constraints. As demonstrated in the figure 2 below, secant lines perform significantly better than tangent lines, whose slope is gradient, in the case of larger noise, particularly for points close to the extreme points. In the work Cheng & Chen (2022), it was shown that for the SOTA models and near SOTA models, the gradient of the inter-layer inputs, i.e., $\frac{\ell}{h_i}$, tends to be zero.

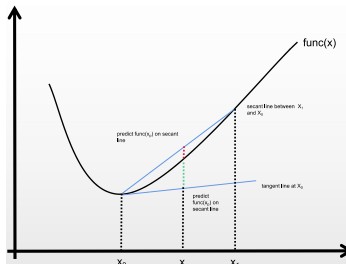

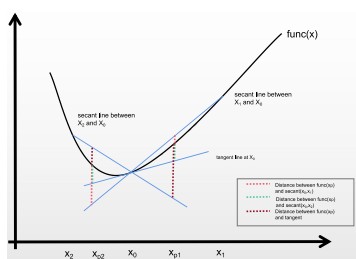

Figure 2: When the predicted point($x_p$) is out of the neighborhood range but not pretty far from $x_0$, secant lines between the $x_0$ and $x_1$ perform significantly better than tangent lines. The choice of $x_1$ is the maximum quantization noise in practice.

Figure 3: Different directions have different secant line in prediction process.

However, the secant line is different in different directions, which is shown figure 3. The direction of the secant line on each feature is different, which leads us to having to compute thousands of different secant lines. To avoid this issue, we only use floor or ceiling quantization methods in a vector to ensure that all data in the vector has the same directions. The more computing method is the content of the work Cheng & Chen (2022).

### 5.1.3 MODEL WEIGHT QUANTIZATION

Based on analyses, we know that the real impact on the fluctuations in the loss function is the error in the quantization of the data between the layers, i.e.,the value of $\epsilon$, and that the error in the weights in a layer almost has no impact on the loss within the "neighbourhood". Ideally, we would expect the quantization, which is leaded by the $\frac{\partial \ell}{\partial h_i}$ is accepted by the concept of the "neighborhood" of weights within layers. In this case, the problem can be set up to easily reach theoretical extreme points, and what is important, the computation cost in this case is low. For the same reason as in the

---

**Algorithm 1:** Change Mixed-Precision Inference Layout Problem into Low computation cost NP hard problem

---

**Input:** Trained Neural Network $M$ which has $n$ layers, Different Quantization Levels
$\quad\quad [q_1, q_2, ..., q_k]$, $error_{max}$, Calibration dataset $D$

**Output:** Price Matrix $P$, Weight Matrix $W$, matrix sizes are n*k

1.Forwark and Backward Network on Calibration Dataset, Collection the distribution of data in inputs on calibration.;

**for** $q_j$ *in* $Q$ **do**

$\quad$ **for** $Layer_i$ *in* $M$ **do**

$\quad\quad$ 2. Compute the $W[i][j]$. (For example, If we quantizate model for the memory cost, $W[i][j]$ is the the memory cost of $Layer_i$ on $q_j$ quantization level);

$\quad\quad$ 3. Compute $\|\epsilon_i\|$ and $scale_{input}$ based on the input of $Layer_i$

$\quad\quad$ 4. Compute $slope_{input} = \|(f(M) - f_{input}(M; scale_{input}; i))/scale_{input}\|$, where $f_{input}(M; scale; i)$ is the loss on the $M$ with the input of $M$'s $Layer_i$ add the $scale_{input} * [1, 1, .., 1]$ for ceil quantization or $scale_{input} * [-1, -1, .., -1]$ for floor quantization.;

$\quad\quad$ 5. Compute $fluc = \|(f(M) - f_{weight}(M; scale_{input}; i))\|$, where $f_{weight}(M; scale_{input}; i)$ is the loss on the $M$ with the weight of $M$'s $Layer_i$ add the $scale_{input} * [1, 1, .., 1]$ for ceil quantization or $scale_{input} * [-1, -1, .., -1]$ for floor quantization.;

$\quad\quad$ **if** $fluc < error_{max}$ **then**

$\quad\quad\quad$ 6.$P[i][j] = slope_{input} * \|\epsilon_i\|/\sqrt{size(\epsilon_i)}$;

$\quad\quad\quad$ 7.Continue;

$\quad\quad$ **end**

$\quad\quad$ 8. Compute $\|\delta_i\|$ and $scale_{weight}$ based on the weight of $Layer_i$;

$\quad\quad$ 9.$P[i][j] = slope_{input} * \|\epsilon_i\|/\sqrt{size(\epsilon_i)} + fluc/scale_{weight} * \|\delta_i\|/\sqrt{size(\delta_i)}$;

$\quad$ **end**

**end**

**return** Price Matrix $P$, Weight Matrix $W$

---

above sections, in practice the noise introduced by quantization often becomes uncontrollable due to the need for speed/model size or due to the quantization needs of the layer input vectors passively. Therefore, the quantization level of this layer must be adjusted to match the weights. In practice, we will first determine whether the noise introduced by the quantization of the model parameters will have a significant effect. The parameter setting in weight quantization is the same as in section 5.1.1. When the noise is significant, it must be corrected by the secant line approach, which is shown in 5.1.2.

## 5.2 PRACTICAL PROBLEM SETTING AND SOLVING

In the above sections, we change the mathematical concepts in Eq. 6 into engineering computations one by one. Then we obtain the following algorithm 1. In this problem transformation, we map through the relationship between quantization level and the noise and the analysis stability of a model on the target data set. To extend the use of the gradient, we use the secant line for linear prediction in the case of larger noise. We convert the quantization problem to a low computation cost NP hard problem, i.e., extended 0-1 knapsack problem as following problem 4:

**Problem 4** *The Low Computational Cost Equivalence Problem of Mixed-precision layout problem*

$$\min \sum_{i=1}^{n} P[i][j], s.t. \sum_{i=1}^{n} W[i][j] < C, j \in [1, k] \text{ and } j \in \mathbf{Z} \quad (8)$$

*where $C$ is the constraint for quantization target. For example, if we quantization model for memory cost, the $C$ is the maximum memory size.*

Problem 4 is the equivalence problem with the problem 3. Compared with problem 3, all elements in problem 4 and algorithm 1 are corresponding to the quantization process parameters or operations. By choosing and selecting at least one $P[i][j]$ in each row of matrix price and one value $W[i][j]$ at the corresponding position in matrix weight, we guarantee that each layer knows which quantization level they should choose to reach the quantization target. Problem 4 is an easy problem to be solved by the branch-and-bound method. This problem can be solved very quickly because it doesn't need inference or training on the original neural network. Instead, it only needs a small number of additions and multiplications.

In traditional concepts, it is generally believed that the less use the quantization bit, the larger the impact on performance, which can be expressed by the value of the loss function. Based on our analysis, i.e., the nature of the 0-1 knapsack problem, we verify that this expectation is correct when quantization introduces small noise. However, it often finds examples of quantization models with better performance with extremely small quantization bits in different papers. This is due to the inability to use the concept of the "neighborhood" of the points for the estimation of the prediction points, i.e., the use of differential or Taylor expansions, which leads to unstable prediction results in the presence of large noise. Given the lack of techniques to perform Fourier expansions on neural networks, the analysis and solution of the problem in the presence of large noise cannot be solved at the current stage.

## 6 EXPERIMENT

In this section, we evaluate the performance of branch and bound method on problem 4. Our objective is to show that branch and bound method based on our proposed problem is better than the full precision model without "fine-tuning" technology, which we use HAWQ-v2 as benchmark.

We use CIFAR 10 dataset. The training dataset is split into calibration and training datasets. Furthermore, the size of the calibration dataset is equal to the test dataset.

Based on the workCheng & Chen (2022), the model with many identity mapping structures has strong noise robustness. Thus we have to choose the model which contain less residual structure. For CIFAR 10, ResNet 20 is close to the SOTA ResNet model(ResNet 110), which mains the layers' input gradient is small. Thus, we choose ResNet 8 and ResNet 14 as our quantization models. Furtherly, to enlarge the $\mathbf{E}\frac{\partial \ell}{\partial h_{i+1}}$, we delete the identity mapping structure in our experimental models.

In quantization practice, quantizing data into INT8 is the most frequently used and ripe choice because current computation devices, like V100GPU, only support INT8, INT16 and INT32 computing in hardware. Thus, we use mixed-precision INT8 and full-precision(FP32) in our experiments. We will compress these model, and our target is to reduce memory consumption by 20 percent. In this experiments, we set $error_{max} = 0.1$ to reduce the computing time.

The loss for ResNet8 with the full precision is 0.3896. The loss for the quantization ResNet8 model is 0.4168 by our method. The loss for the quantization ResNet8 model is 0.4418 by HAWQ-v2. The loss for ResNet14 with the full precision is 0.3634. The loss for the quantization ResNet8 model is 0.3892 by our method. The loss for the quantization ResNet14 model is 0.3964 by HAWQ-v2. The experimental results match our analysis.

## 7 CONCLUSION

In this paper we propose the algorithm that change mix-precision problem, i.e. problem 2 into a low computation cost NP hard problem, i.e., problem 4. Based on the total differential, we show that the fluctuation of loss function is mainly related to the gradient of layer's input and the noise of quantization. We map mathematical concept into practical operations. After gain the NP hard problem, branch and bound methods to solve the problem efficiently and our quantized model is better than the model from HAWQ-v2 with the same quantization limitation.

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
