# OpenReview forum: "Mixed-Precision Inference Quantization: Problem Resetting, Mapping math concept and Branch\&bound methods"
_ICLR.cc/2023/Conference — Submitted to ICLR 2023_

### Official Review · Reviewer_nDrD · 2022-10-23

**Confidence:** 4
**Correctness:** 2
**Technical Novelty And Significance:** 2
**Empirical Novelty And Significance:** 1
**Recommendation:** 3

**Clarity, Quality, Novelty And Reproducibility:**

The paper is difficult to track and does not follow the standard of academic writing. For example:
- In the introduction, there is no citation to justify the statements.
- Section 3 is cumbersome: for ICLR audience, these background should not take more than half-page.

The empirical study is casual. There is a lack of presentation of the experimental setting, the central hypothesis, and the justification of the selection of the baseline (there are many works in neural network quantization, a more exhausted comparison should be included). Additionally, some claimed contribution from the introduction is not verified by the experiments, e.g., based on the description, it is unclear to me why the proposed method "has clear interpretability", what does interpretability refer to?

There is a lack of code release for reproducibility or a plan to do so.

**Strength And Weaknesses:**

Weakness:


- The paper is not well-written.

- The empirical study is casual.

**Summary Of The Paper:**

This paper propose a neural network quantization method by mapping the mixed-precision layout problem into a traditional NP-hard problem, and solve the problem by some efficient implementation of relaxed solution to these NP-hard problem.

**Summary Of The Review:**

This paper is difficult to track, and the claimed contribution is not supported by the experiments.

---

### Official Review · Reviewer_hSyo · 2022-10-24

**Confidence:** 4
**Correctness:** 1
**Technical Novelty And Significance:** 1
**Empirical Novelty And Significance:** 1
**Recommendation:** 3

**Clarity, Quality, Novelty And Reproducibility:**

* Clarity:

  * I found the paper extremely hard to follow, as English is poor and lacks a top-down logical flow. Examples:
    * "we will show that how to mixed use round up/round down" (page 3)
    * "to describe the conclusion for the sequential neural network is easily described" (page 4)
    * "because of the natural of quantization is the trade of the performance and computation resource use" (page 4)
    * "equation 7 is purity mathematical concept" (page 6)
  * The authors do not clearly state their contribution.
  * Mathematical notation and rigour are lacking:
    * Equation (5): this is a linear approximation (so the equal sign should be replaced by an approximation sign).
    * Equation (6):
      * $E$ is not defined, nor is $e$
      * $l$ is the loss for training example $j$, it should take $(x_j, y_j)$ as arguments
    * Equation (8):
      * What is the minimum taken over?
      * Is there a separate optimization problem here for each j? Or is j a function of i, that is, for each $j$
      we look for a different $i$?
      * Algorithm 1: Whenever a layer is quantized, it is not clear whether the previous layers remain full precision
      or not. I would assume yes, but this is not specified.
  * Some equations/formulas do not look correct to me:
    * Equation (6): Assuming $e$ consists of a vector of $\epsilon_i$ and $\delta_i$ and $E$ is
    the entire real domain (hard to tell, as $E$ is undefined), then the objective on the left
    hand side is linear in $e$ and unbounded in general (one can set $\epsilon_i$ to be arbitrarily large in absolute
    value and to have sign opposite to the sign of
    $\sum_{i=1}^{n} \sum_{(x_j, y_j) \in \mathbb{D}} \frac{\partial l}{\partial h_{i+1}}$
    and a similar expression exists for $\delta_i$).
    * Algorithm 1: The slopes on lines 3 and 5 are taken in absolute value. If the validation loss decreases (which can
    sometimes happen in practice as quantization can act as a regularizer), there will still be a positive price
    incurred on line 9.
    * Equation (7): What variable is the minimum taken over ($w$, $\epsilon$)?
    * Why is there an "or" in equation (7)? Are these two alternative minimization problems?
  * The authors mention "What is more, in the model, which consists mostly of identity mapping [...]". What do the
  authors refer to by "mostly"? Is this a reference to the residual blocks in ResNet?

* Novelty:

  * It is not clear what the contribution of this paper is. It mostly seems to propose an additive cost model, while the
  rest of the algorithm simply relies on ACIQ and a linear approximation of the loss. It is fairly trivial, and no
  studies are carried out to show that this assumption holds in practice. Hence, I believe there little technical
  novelty.

* Quality:

  * Given the very limited experimental scope and that no validation of the additive model and/or linear
  approximation hypotheses is considered, I do not think this work does not meet the scientific quality standards for
  publication.

* Reproducibility:

  * Since there are so many unclear points with respect to the method, reproducibility is hardly possible, in my
  opinion. The authors also do not mention the intention to release any public code.

**Strength And Weaknesses:**

* Strengths:

  * In cases where the independence assumptions hold, the proposed algorithm should be computationally efficient.

* Weaknesses:

  * The paper lacks clarity and notational rigour (please see notes below on clarity).
  * The assumption of additive accuracy price and computational cost model may not hold in practice and the authors
  do not conduct a study on this. The total accuracy degradation is likely not a linear function of layer-wise accuracy
  losses, as layers interact via feature maps. Furthermore, if one considers running time as computational cost,
  inter-layer interactions arise from cache locality and bandwidth effects, leading to non-additive models.
  * The authors simply report the loss (not clear what loss is being reported) and compare against a single baseline,
  on a single (and not very challenging) classification dataset with two network architectures. This is hardly
  convincing. No ablation studies are shown and the additive model hypothesis is not validated.

**Summary Of The Paper:**

The paper lacks clarity, so I reproduce here my best understanding of its contents.

The paper proposes to search for the optimum bit width (or more generally, numerical precision) for each layer in a
neural network during post-training quantization. To this end, the authors assume quantization induces an additive
price model -- <i>i.e.</i> each layer's quantization incurs an accuracy cost with respect to the full precision version,
independent on the quantization parameters of the other layers -- and an additive cost model -- i.e. layer computational
costs are independent on each other. Under these assumptions, finding the optimal bit widths can be cast as a 0-1
knapsack problem. The price for quantizing each layer is estimated via a first-order linear approximation of the loss
function with respect to the quantization-induced noise. The only experiment the authors report is the classification
loss (not clear which loss is being reported) for ReseNet8 on CIFAR-10.


**Summary Of The Review:**

This work does not seem to be technically novel. The presentation style and mathematical rigour needs to be
greatly improved. To me, there are many unclear points and questions regarding correctness. Empirical support for the
method and hypotheses is lacking (and I would expect that the additive model does not hold in practice). Overall,
I do not think this work meets the minimal standards for publication at ICLR.

---

### Official Review · Reviewer_y7Tw · 2022-10-24

**Confidence:** 3
**Correctness:** 3
**Technical Novelty And Significance:** 3
**Empirical Novelty And Significance:** 2
**Recommendation:** 5

**Clarity, Quality, Novelty And Reproducibility:**

The paper proposed a novel framework to solve the mixed-precision inference quantization problem, it’s clear and well written.

**Strength And Weaknesses:**

Strengths:
1. Based on the total differential calculation, the paper proposed a theoretical framework incorporating bot the quantization error and data type conversion error.
2. The paper also proposed the secant line method which fixed the large approximation error issue.
Weaknesses:
1. The method mainly focuses on the models which contain less residual structure. However, most of the popular models contain the residual structure such as transformer et al.
2. The experimental results are not enough to show the strengths of the proposed framework: it only test simple architectures. It will be much better to show more results on more models, more baselines and more statistics such as accuracy and so on.


**Summary Of The Paper:**

In  this paper,  the mixed-precision layer problem is formulated as a traditional NP hard problem and the problem can be solved by low cost methods without fine-tuning. The experimental results show that the proposed method is better than the current SOTA method.


**Summary Of The Review:**

To solve the challenging mixed-precision inference quantization issue, this paper proposed a theoretical framework to convert the original problem into a traditional NP hard problem, the derivation and writing is clear. However, the experimental results are limited to show the benefits on popular neural networks.

---

### Decision · Program_Chairs · 2023-01-20

**Decision:**

Reject

**Justification For Why Not Higher Score:**

This is a clear cut case: all reviewers recommended reject, and the authors did not respond.

**Justification For Why Not Lower Score:**

N/A

**Metareview: Summary, Strengths And Weaknesses:**

This paper designs a mixed precision method for neural networks and tests it on a few neural networks. Though the direction of improving neural networks efficiency via better quantization is important, the reviewers and myself found many issues, such as:
1. Lack of rigor and clarity (see Reviewer hSyo for examples).
2. The results reported in the evaluation section are very limited, and partial, and with many missing details (e.g., which type of loss is compared?).
3. The novelty of this paper is not clear, and how it compares to previous similar methods. For example, using the assumption of additive loss for each layer, so we can plan this as integer programming problem was done before [A,B]. So why were the results here only compared against HAWQ-V2 but not [A] and HAWQ-V3 [B]?

Therefore, all reviewers recommended reject, and the authors did not respond.


[A] Hubara et al., Accurate Post Training Quantization With Small Calibration Sets, ICML 2021

[B] Yao et al., HAWQ-V3: Dyadic Neural Network Quantization, ICML 2021